# Layered polymer-perovskite composite membranes for ultraflexible fatigue-tolerant optoelectronics

Yalu Li[1], Can Zou[1], Da Liu[1], Qing Li[1], Yan Zhu[1], Miaoyu Lin[1], Sihan Zeng[1], Zhanpeng Wei[1], Xinyi Liu[1], Yichu Zheng [2], Yu Peng [1], Yu Hou [1] ✉, Hua Gui Yang[1] ✉ & Shuang Yang [1] ✉

Flexible integration of perovskite materials has driven diverse applications, from wearable detectors, portable energy systems to foldable displays. However, due to the intrinsic brittleness of perovskite, mechanical strain inevitably causes the degradation and variation of electronic performance of the devices. Here, we establish a periodic multilayered polymer-perovskite membrane that showcases plastic-like mechanical behaviors of small Young's modulus (5.41 GPa) and bending tolerance (radius of 0.5 mm), yet retains the perovskite's carrier transport capacity ($\mu\tau$ product of $1.04 \times 10^{-4}$ cm$^2$ V$^{-1}$). The mechanistic study shows that the formation of bicontinuous perovskite-polyimide structure in the membrane accounts for the carrier transport and load transfer functions, respectively, thus unifies paradoxical mechanical and electronic properties. Using a lateral device configuration, X-ray detector based on the membrane delivers a high X-ray sensitivity of 8380.80 μC Gy$_{air}^{-1}$ cm$^{-2}$, and withstands 30,000 repeated bending cycles under a bending radius of 1.5 mm without notable performance degradation.

Soft, lightweight, and electrically robust semiconductors are cornerstones for next-generation flexible electronic devices that function as a nontrivial platform for wearable sensors, soft robotics, and implanted devices[1,2]. Organic materials with inherent flexibility have been recognized to be particularly suitable for flexible devices, but they often suffer from unsatisfactory electronic performance caused by the hopping transport behavior[3]. On the contrary, inorganic semiconductors, like GaAs and silicon, can achieve long carrier diffusion lengths, yet the ionic or covalent nature renders them inherently brittle and susceptible to fracture upon structural deformation[4]. Therefore, a fundamental trade-off occurs between mechanical flexibility and electronic property in most existing materials.

Lead halide perovskites are emerging semiconductors with high absorption coefficients, long carrier diffusion lengths, tunable bandgaps, and defect tolerance, which revolutionized the research field of optoelectronics, including solar cells, light-emitting diodes, and radiation detectors in the past decade[5–9]. Unlike conventional semiconductors that necessitate vapor or vacuum processing, the solution fabrication of perovskite offers a technical route to integrate these brittle materials with soft organic matters to gain moderate flexibility. For instance, perovskite thin films can be deposited on bendable substrates, e.g., polyethylene terephthalate (PET) and parylene, which serve as the mainstream building blocks for pliable devices[10]. However, these devices still suffer from poor bending tolerance and delamination problems[11,12]. In recent years, chemical and physical approaches, including molecular cross-linking[13,14], interface strengthening[15,16], perovskite composition engineering[17,18], polymer blending[19], and nanostructure patterning[20,21], have been proposed to minimize the mechanical fatigue of perovskite films. Nevertheless, few of these progresses achieved both super-flexibility and good electronic

[1]Key Laboratory for Ultrafine Materials of Ministry of Education, Shanghai Engineering Research Center of Hierarchical Nanomaterials, School of Materials Science and Engineering, East China University of Science and Technology, Shanghai, China. [2]School of Mechatronic Engineering and Automation, Shanghai University, Shanghai, China. ✉e-mail: yhou@ecust.edu.cn; hgyang@ecust.edu.cn; syang@ecust.edu.cn

properties that can deliver a stable output signal under mechanical deformation and angular change. According to Hooke's law and the series and parallel spring model, the strain distribution of a deformed substrate is correlated with the stiffness of certain localized regions, and that is, the incorporation of soft matrix into perovskite films should be useful to receive the strain experienced by the bulk perovskites[22–24]. However, both the capacities to transport charges and dissipate mechanical stress, as a prerequisite for the flexible electronic application, cannot be readily realized in typical composite architectures[25].

Here, we report the design and spray fabrication of strain-insensitive, intrinsically flexible composite membranes that consist of periodic soft-hard bicontinuous stacking layers. In particular, the microscopic geometry is engineered based on three considerations: mechanical strain and deformation concentrate on the soft continuous polyimide (PI) phase when exposed to repeated bending or folding; strong interfacial interaction to stabilize the hetero-interface and facilitate load transfer; photocarriers efficiently diffuse/drift along the lateral continuous perovskite planes for radiation detection function. Lateral device based on the composite membrane achieves a high X-ray sensitivity of 8380.80 $\mu C\,Gy_{air}^{-1}\,cm^{-2}$ under an electric field of $500\,V\,cm^{-1}$, and enables large-area imaging and wearable dynamic detection applications. More importantly, the detection performance of the device has been fully sustained over 30,000 bending cycles at an extreme bending radius of 1.5 mm, and is also tolerant to 2000 cycles of wrinkling tests with micro-wrinkle radius <200 $\mu$m, which allows flexible perovskite devices to be applied in a broader scope of applications.

## Results

### Fabrication of free-standing composite membrane

We fabricated the composite membrane through a layer-by-layer spray-coating process of polymer and perovskite co-dissolved in N,N-dimethylformamide (DMF)/dimethyl sulfoxide (DMSO) solution onto glass substrates (Fig. 1a, d; see details in "Experimental section"). PI utilized in this work, as one of the superior-performing engineering plastics with exceptional thermal stability, mechanical strength, and chemical resistance (Supplementary Fig. 1), shows solution miscibility with many common perovskite compositions[26], such as $CsPbBr_3$, $MAPbBr_3$, $FAPbI_3$, and $Cs_2AgBiBr_6$ (MA is methylamine and FA is formamidine) for spray deposition. We also found that other polymers, like polyvinyl pyrrolidone (PVP), which exhibit both miscibility with the perovskite precursor solution and the capability of gelation, are compatible with this fabrication protocol (Supplementary Fig. 2). During the deposition procedure, the inorganic and organic constituents underwent fundamentally different processes upon solvent evaporation: the perovskite phase rapidly nucleated and grew once oversaturation[27], while the macromolecule chains entangled with each other and manifested as gelation before complete desolvation[28,29]. In contrast to typical ternary systems with two different polymers that are usually characterized by liquid–liquid demixing to polymer-rich and polymer-poor phases, perovskite crystals are directly generated and dispersed in the concentrated polymer gels with residual solvent molecules, followed by the shrinkage of the polymer network[30]. This process can be described by the trajectory of the compositional path of the mixed solution within the ternary phase diagram (Supplementary Fig. 3). When deposited on a flat substrate, the spreading of precursor droplets allowed the enrichment of perovskite phase at the middle regime of the gelatinous products (Supplementary Fig. 4), and gradually self-organized into layered-like microstructures of soft PI and hard perovskite phases via spray-coating (Fig. 1b), wherein the connectivity of biphase was predominately determined by the ratio of the two components. We also fabricated the composite membrane through the spin-coating process, but the layered-like microstructures

disappeared, which should be closely related to the dispersed solvent evaporation and crystallization kinetics (Supplementary Fig. 5).

In principle, the photoelectronic and mechanical properties of the membrane are generally defined by the continuity of both polymer-perovskite phases: the perovskite phase establishes semiconducting percolation networks with its photoelectronic properties undergoing a dramatic change as the mass fraction ($f$) of perovskite filler phase approaches a critical value ($f_c$), i.e., the percolation threshold[31]; the continuous polymer phase accommodates the mechanical strain and dissipates local stress as a soft supporting framework (Fig. 1c)[24]. In these architectures, the semiconductivity and flexibility meet a trade-off in most occasions, but they may coexist in a membrane with a bicontinuous phase. By adjusting the mass fraction of perovskite to be 40–80%, we assembled composite membrane with layered co-continuous layered phases in three dimensions (details about the structures will be discussed later), which leverage the flexibility and carrier collection (Fig. 1h). One notable advantage of the spray-coating fabrication approach is the broad tunability in membrane size, shape and thickness, which can be delicately modulated by the substrate size, geometry and printing cycles, respectively[32]. For instance, the membrane can be spray-deposited on many irregular substrates, like disk-, heart- or flower-shaped ones, that can not be achieved via spin-coating or doctor-blading techniques (Supplementary Fig. 6). We systematically investigated the processing parameters of annealing temperature, annealing time, and the number of spray cycles, and found that the microstructure of composite membrane can be broadly tailored by these parameters (Supplementary Fig. 7). Compared with most perovskite films that strongly attached to underlying substrates, we found that the composite membrane with over 100 $cm^2$ area can be readily peeled off as a free-standing membrane due to the weak PI–glass interaction (Fig. 1e and Supplementary Fig. 8). These membrane with layered structure can withstand multiple mechanical deformations, such as twisting, bending to a radius of 0.1 mm and puncturing by a sharp needle (Fig. 1f and Supplementary Fig. 9). The durable resilience to various damage scenarios arises from the energy-dissipative PI framework, which would circumvent the mechanical damage experienced in daily usages. In the following paragraph, we exemplify the structural and physical properties of PI-perovskite membranes, and henceforth refer to it as the composite membrane if not specified.

The layer-by-layer assembly of the PI-perovskite produced periodic stacking laminates of continuous $CsPbBr_3$ and PI networks (Supplementary Fig. 10). Cross-sectional scanning electron microscope (SEM) images of the composite membrane in Fig. 1g demonstrate the in-plane connectivity of both phases with an average unit layer thickness of about 150 nm. The elemental distribution of the stacked perovskite/polymer planes in the layered microstructure is then visualized by energy dispersive X-ray spectroscopy (EDX) mapping (Supplementary Fig. 11) and time-of-flight secondary ion mass spectrometry (ToF-SIMS) analysis (Supplementary Fig. 12). Furthermore, the prior precipitation of perovskite within the gelatinous matrix yielded smooth top surface enclosed by compact PI phase with the surface-root-mean square of 11.72 nm over a scanning area of 5 × 5 $\mu m^2$ (Supplementary Fig. 13), which is comparable to that of perovskite films in high-performance solar cells[33]. To enable closer inspection, the membrane was ball-milled and ultrasonicated in toluene solution to obtain fragment samples. Transmission electron microscopy images of the fragments in Supplementary Fig. 14 reveal an immiscible yet closely contacted biphase heterostructure that can be persistent upon severe damage in most observed regions, indicative of the solid interconnection at the two-phase interface. The interfacial interaction was subsequently elaborated by Fourier-transform infrared spectra (FT-IR), where the shift of C–N–C stretching vibration of 1374 $cm^{-1}$ implicates the chemical bonding of the nitrogen donors in the amide moiety to surface metal centers in perovskites (Supplementary Fig. 15). Spherical

aberration-corrected high-angle annular dark-field scanning transmission electron microscopy images in Fig. 1g show no obvious defects or lattice deformation, manifesting the high crystallinity and phase-purity of the perovskite phase.

## Mechanical property and energy dissipation mechanisms

Although the perovskite is incorporated in the composite membrane, it retains the physical elasticity and structural reversibility of the PI phase. To prove it, we characterized the mechanical properties of the membranes by uniaxial tensile test and nanoindentation measurement (Fig. 2a). Representative stress–strain curves obtained from the tensile test in Fig. 2b showcase a tensile strength ($\sigma$) of 27.44 MPa and an elongation at break ($\varepsilon_{EB}$) of 4.73%, respectively, for the composite membrane. On the contrary, pure ionic perovskite

would fracture immediately once stretching, e.g., MAPbI$_3$ films broke at 1.17% $\varepsilon_{EB}$ in micro-scale tensile tests[34]. These observations pinpoint that the in-plane mechanical character of the membrane would be drastically changed from fragile ionic materials to stretchable and tough ones that are inherited from the PI frameworks. The Young's modulus ($E$) and hardness ($H$) values were then derived from load ($P$)–displacement ($h$) curves following the Oliver-Pharr method (Fig. 2c, d)[35,36]. The mechanical responses of the PI membrane and pure perovskite on the PET substrate were also included as a comparison. During a loading-unloading cycle, the hysteresis loop of the composite membrane shows resemblance to that of PI, resulting in hardness of 0.416 GPa for PI membranes, 0.461 GPa for composite membranes, and 0.646 GPa for pure perovskite films, respectively. Similar trends occurred in Young's modulus of these membranes

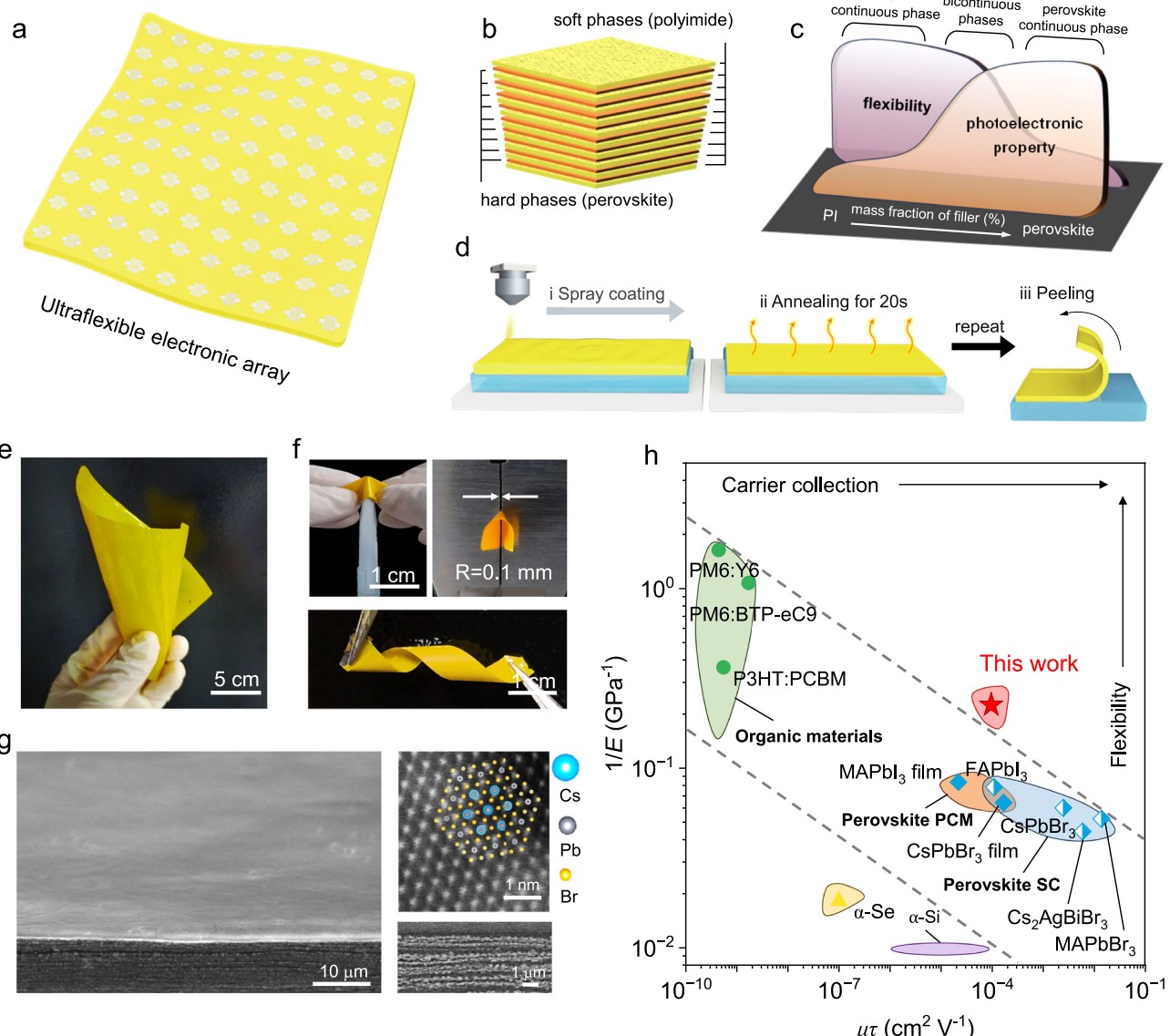

**Fig. 1 | Fabrication of composite membranes. a** Schematic of free-standing composite membrane for ultraflexible electronic application. **b** Self-organized layered microstructures constituting bicontinuous periodic polymer and perovskite phase. **c** Dependence of mechanical flexibility and photoelectric property of the composite membrane as a function of filler (perovskite) mass fraction ($f$). **d** Scalable assembly of composite membranes via layer-by-layer spray-coating technique. **e** Photograph of a freestanding large-area ($10 \times 10$ cm$^2$) composite membrane under folding state. Scale bar is 5 cm. **f** Photographs of composite

membranes showing ultraflexibility by twisting, poking and bending. Scale bars are 1 cm. **g** Oblique-view and cross-sectional SEM images and HAADF-STEM image of the composite membrane. Inset is the atomic model structure of CsPbBr$_3$ perovskite. Blue: cesium atoms; gray: lead atoms; yellow: bromine atoms. Scale bars are 10 μm, 1 μm, and 1 nm, respectively. **h** Summary of the mobility-lifetime product ($\mu\tau$) and Young's modulus ($E$) for different materials. PCF and SC refer to polycrystalline film and single crystal, respectively.

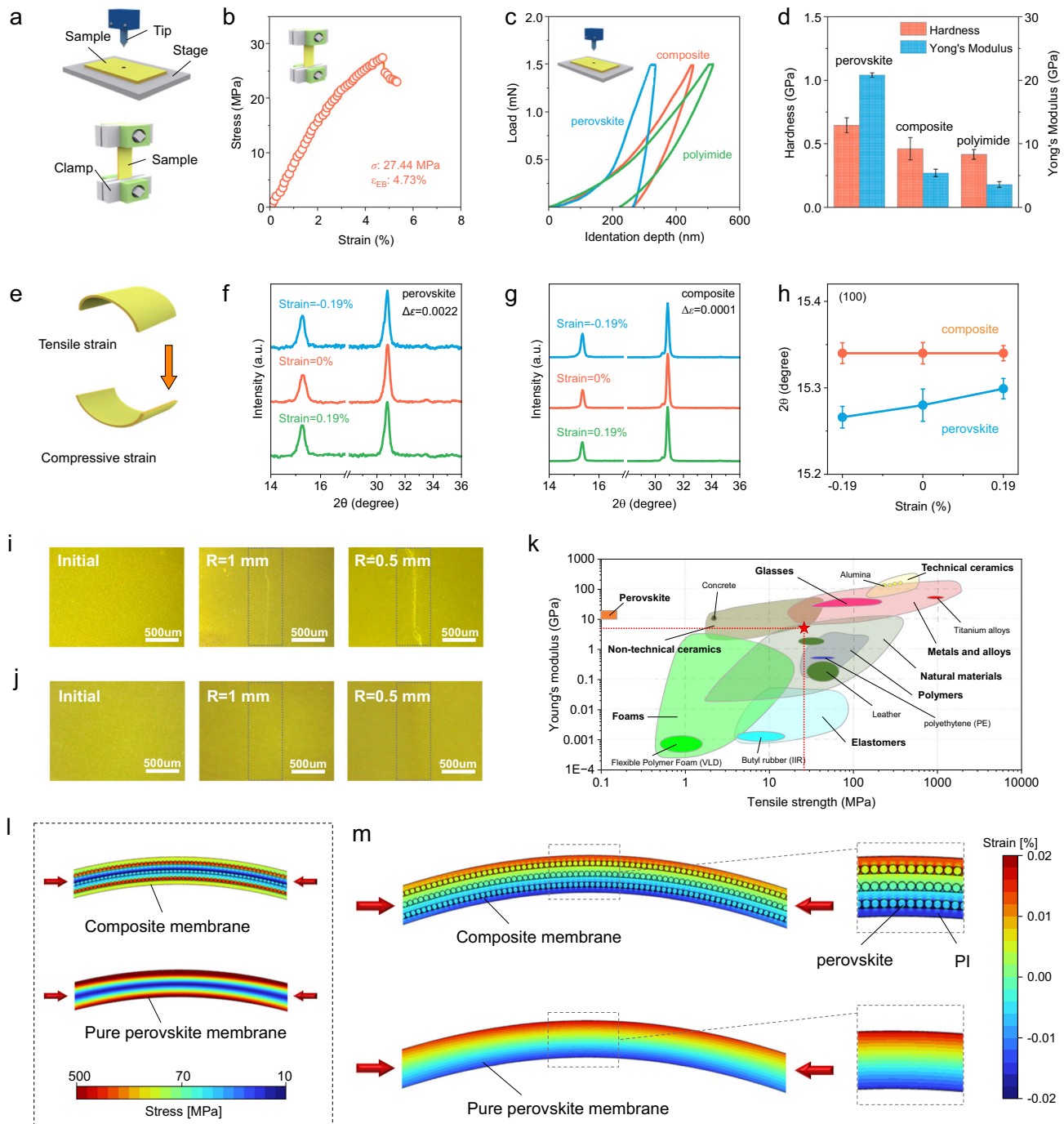

**Fig. 2 | Mechanical properties and strain tolerance. a** Experimental setup of the nanoindentation and uniaxial tensile tests. **b** Tensile stress–strain curve of the composite membrane. **c** Indentation load–depth curves of perovskite membranes on PET substrates, PI membranes, and composite membranes in a load-controlled mode. **d** Hardness ($H$) and Young's Modulus ($E$) of samples measured by nanoindentation test. Error bars represent the standard deviation (SD). **e** Illustration of the XRD measurement of samples upon the tensile strain to the compressive strain. XRD patterns of (**f**) perovskite membranes and (**g**) composite membranes under applied strain of 0.19%, 0%, −0.19%. **h** XRD peaks of (100) plane as a function of applied strain. Error bars represent SD. Optical microscope images of (**i**) perovskite membranes and (**j**) composite membranes after bending at different radii. Scale bars are 500 μm. The gray area was the bending zone. **k** Ashby plot representing Young's modulus against the tensile strength. Composite membranes reported in this paper are highlighted by the pentagram. There is only data on Young's Modulus about perovskite because it is difficult to fabricate free-standing perovskite membranes in uniaxial tensile tests. Finite element simulations for (**l**) stress and (**m**) strain distribution of perovskite membranes and composite membranes.

where the $E$ values of PI membrane, composite membrane, and perovskite film were measured to be 3.585 GPa, 5.411 GPa, and 20.828 GPa, respectively. In spite of the presence of perovskite phases along the in-plane orientation, we speculate that the applied strain perpendicular to the membrane in the nanoindentation experiment should be primary located on the soft polymer phase, rather than the bulk perovskite crystals, similar to natural layered materials such as nacre[37] and tooth[38], and dissipate the energy by the chain dynamics, like rotation, elongation, sliding etc[39]. Figure 2k demonstrates the comparison of the tensile strength and the Young's

modulus between different reported materials and composite membranes.

Since strain is embodied as the variation of lattice parameter in crystalline materials, XRD measurements are conducted as a convenient and effective technique to estimate the global strain of the perovskite phase upon deformation. Tensile and compressive strains were applied to PET substrates as the supporting layers for all films (Fig. 2e, the PET substrates are hidden from view). According to the positive Poisson's ratio in perovskites, a tensile loading on the perovskite film would stretch the lattice spacing along the in-plane direction; the lattice plane perpendicular to the substrate would be compressed simultaneously, and vice versa[40]. XRD $\theta$–$2\theta$ scan profiles in Fig. 2f–h display that the (100) diffraction peaks of composite membranes are essentially stationary upon bending, whereas the perovskite films experience an opposite shift of (100) diffraction peaks from tensile to compressive strains. The calculated bending strain difference ($\Delta\varepsilon$) under two strained states of the composite membranes is only 0.0001, which is remarkably lower than the $\Delta\varepsilon = 0.0022$ of the perovskite films (Supplementary Notes 1, 2). An identical phenomenon was captured in the Raman spectra: the wavenumber of the vibrational modes of lead bromide octahedron, as a measure of unit cell dimensions, shows a pronounced shift in the pure perovskite films under strained conditions (Supplementary Fig. 16)[41,42].

When subjected to bending radius down to 0.5 mm, the composite membranes can be fully recovered to the initial state without any observable damage (Fig. 2j and Supplementary Fig. 17). By contrast, the brittle perovskite films on PET substrate underwent apparent plastic deformation with nonrecoverable cracks at bending radius of 1 mm (Fig. 2i). Aforementioned results have evidenced negligible strain at bulk perovskites in the bended membrane, we conducted finite element simulations to further deconvolute the underlying mechanics in terms of the pure perovskite and hierarchical composite structures (see Supplementary Note 3 for details). For films with the same bending angle of 50°, the composite membrane experienced a lower stress value than the perovskite membrane (Fig. 2l), in which the maximum tensile stress on the perovskite particle was less than half of the value of the pure perovskite film. Owing to the layer-by-layer structure, the stress can be effectively transferred by shear deformation of the interface between the hard and soft systems[43,44], resulting in the enhancement of toughness and flexibility of the composite membrane[45]. In pure perovskite film, the stress is concentrated on the surface region with the largest deformation. In principle, for a given bending deformation, the stiff perovskite phase resists structural deformation, while the PI deforms more to accommodate the strain. This phenomenon is further visualized in Fig. 2m, where both tensile and compressive strains are concentrated in the PI layer, especially near the perovskite particles. According to the amorphous and soft nature of the PI phase, the strain energy can be elastically recovered and/or dissipated via a variety of dissipative routes, e.g., molecular rotation, chain elongation, chain sliding or alignment, which thus explains the mechanical robustness of the composite membrane[46]. In comparison, a large strain zone developed at the top surface in the CsPbBr$_3$ film, corresponding to a large tensile stress field where cracks could generate and propagate.

## Anisotropic percolative charge transport and photoelectric properties

Electronic conduction in multiphase composite materials generally complies with the well-known percolation theory for charge transport, in which the connectivity of conductive regions determines the overall electrical property[31]. When scrutinizing cross-sectional SEM images of composite membranes with varied filler (perovskite) mass fraction ($f$), typical percolation behavior of perovskite phase can be unambiguously observed, yet with preferential connection along different directions (Fig. 3a). At a low perovskite mass fraction of 30 wt%,

discrete perovskite grains appeared in the layered-like polymer matrix without notable connection along both the vertical and parallel direction to the membrane. Interestingly, when 50 wt% perovskite is employed, the perovskite phase is linked along the in-plane direction. Further increasing of perovskite mass fraction would narrow the out-of-plane spacing of organic phase and eventually engender the full contact of perovskite phase along the out-of-plane direction at 90 wt% of perovskite mass fraction (Supplementary Fig. 18). Unlike most identified percolative systems featuring isotropous connection events, the percolation behavior of random-shaped perovskite grains in this case is fairly dependent on the orientation stemmed from the unique layer-by-layer fabrication process, and we termed it as an "abnormal anisotropic percolation" behavior.

The electrical resistivities of lateral and vertical devices with different mass fractions of perovskite are summarized in Fig. 3b, c. The charge transport in the membranes can be divided into three regions: (1) a nonconducting plateau with resistivity close to pure insulating PI phase; (2) a critical region with dramatic transition in resistivity; (3) a semi-conducting region with resistivity approaching that of pure perovskite phase. Combined with the schematic of the membrane structure (Fig. 3h), the range of carrier motion (orange area) extends from within isolated particles to lateral perovskite films, and finally within the whole bulk of the membrane. It can be seen that the evolution of resistivities in both lateral and vertical devices shows consistency with that in the membrane morphology. In Fig. 3d, e, we show the resistivity ($\rho$) of the lateral and vertical devices as a function of perovskite mass fraction at the conduction state. When the perovskite forms a continuous percolating path throughout the matrix ($f > f_c$), a least-squares analysis of the fits shows a clear boundary between the highest insulating point and the lowest conducting point (Supplementary Note 4). Plots of log ($\rho$) versus log ($f - f_c$) in the insets provide the critical thresholds ($f_c$) and critical exponent ($t$) in these devices[47]: the $f_c$ values of lateral and vertical devices are 39.8% and 50.2%, respectively, which further proves the lateral conduction first. The emergence of two distinct percolation thresholds was also reflected in the anisotropic resistivity ratio, defined as the resistivity ratio of the vertical device to the lateral device at the same mass fraction (Supplementary Fig. 19).

We operated steady-state photoconductivity measurements to derive the mobility-lifetime ($\mu\tau$) product, as an important figure-of-merit for evaluating carrier collection efficiency[48], of the membrane with 65 wt% perovskite (Fig. 3f). The derived $\mu\tau$ product of the lateral device is $1.04 \times 10^{-4}$ cm$^2$ V$^{-1}$, which is approximately three orders of magnitude larger than the vertical one ($2.78 \times 10^{-7}$ cm$^2$ V$^{-1}$) because of the interruption of out-of-plane carrier transport by nonconductive PI layers. The large resistance observed in the lateral device normally pairs with a low dark current, which is beneficial for detection applications (Fig. 3g). In addition, the UV–vis absorption spectra and time-resolved PL spectra show that the presence of PI phase does not affect the optical bandgap and charge carrier recombination of the perovskite phase (Supplementary Figs. 20, 21).

## X-ray detection performance and imaging

The multiple-layered superstructure with bicontinuous phase in combination with strong interfacial coupling and high perovskite crystallinity offers the possibility to construct electronic devices with both mechanical flexibility and in-plane photo-electric response. We therefore fabricated a flexible perovskite X-ray detector and evaluated its sensitivity and detection limit based on a lateral device configuration with the best perovskite mass fraction of 65%, which has both excellent photoelectronic properties and mechanical flexibility. We placed a Pb mask on top of the device for accurately defining the effective area, and monitored the photocurrent under X-ray dose rate varying from 0.74 $\mu$Gy$_{air}$ s$^{-1}$ to 0.03 $\mu$Gy$_{air}$ s$^{-1}$ at an electric field of 500 V cm$^{-1}$. Current-time plots in Fig. 4a showcase that the current signal of the detectors is

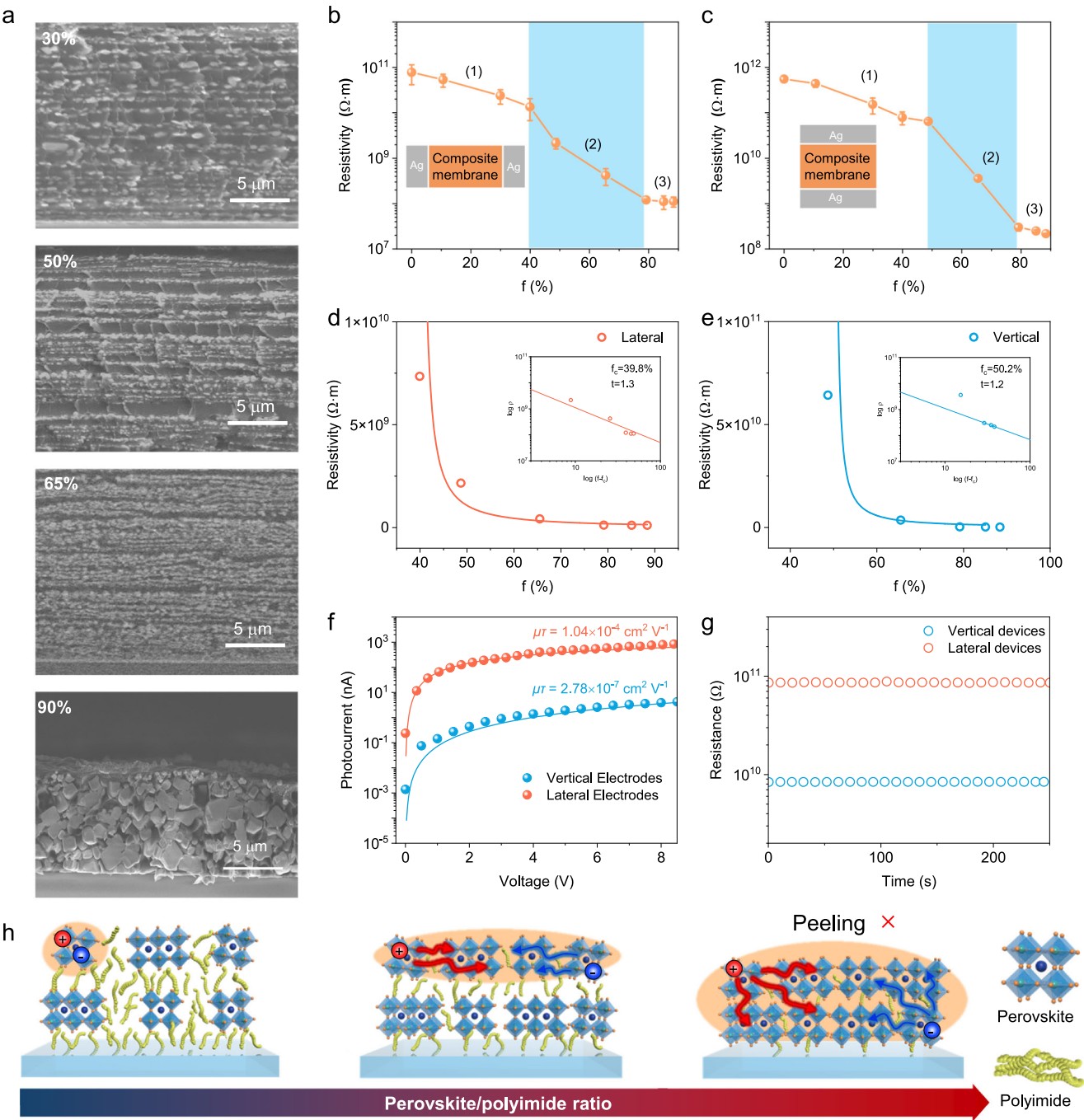

**Fig. 3 | Structure-dependent anisotropic electronical transport. a** Cross-sectional SEM images of composite membranes with different perovskite mass content. Scale bars are 5 μm. Resistivities of (**b**) lateral and (**c**) vertical devices as a function of the mass fraction of perovskite. Insets show the configuration of the measured devices. Membranes maintain a lamellar structure in the blue area. Error bars represent SD. Resistivities of (**d**) lateral and (**e**) vertical devices as a function of the mass fraction of perovskite. Solid lines in both graphs are the fitting of experimental data based on the percolation model. Insets show log-log plots according to the best fits for percolation threshold ($f_c$) and critical exponent ($t$). **f** Photoconductivity measurements of lateral and vertical devices with perovskite mass fraction of 65%. **g** Resistance measurements of lateral and vertical devices with a perovskite mass fraction of 65%. **h** Representative structure of the composite membrane with varied perovskite/PI ratio, resulting in the change in anisotropic charge transport and mechanical strippability. The orange area represents the delocalization region of charge carriers.

dominated by X-ray intensity, and rapidly responds to its on/off switching. The photocurrent increases significantly with increasing bias voltage, compared to the almost negligible increase of dark current (Supplementary Fig. 22). The sensitivity of the X-ray detector, calculated by the linear fitting of the dose rate dependent response current[49], achieves 8380.80 μC Gy$_{air}^{-1}$ cm$^{-2}$ under an electric field of 500 V cm$^{-1}$. Due to the large resistance and low noise of

the device, the detection limit at a signal-to-noise ratio (SNR) of 3 is estimated to be as low as 26.37 nGy$_{air}$ s$^{-1}$ (Fig. 4b), which is over 200 times lower than that required for practical medical diagnosis (5.5 μGy$_{air}$ s$^{-1}$)[48]. Moreover, the outstanding performance exceeds that of most reported flexible X-ray detectors[50]. Furthermore, the reliability of the detector was examined under three conditions, i.e., continuous X-ray exposure at 15.41 μGy$_{air}$ s$^{-1}$ for 40 min

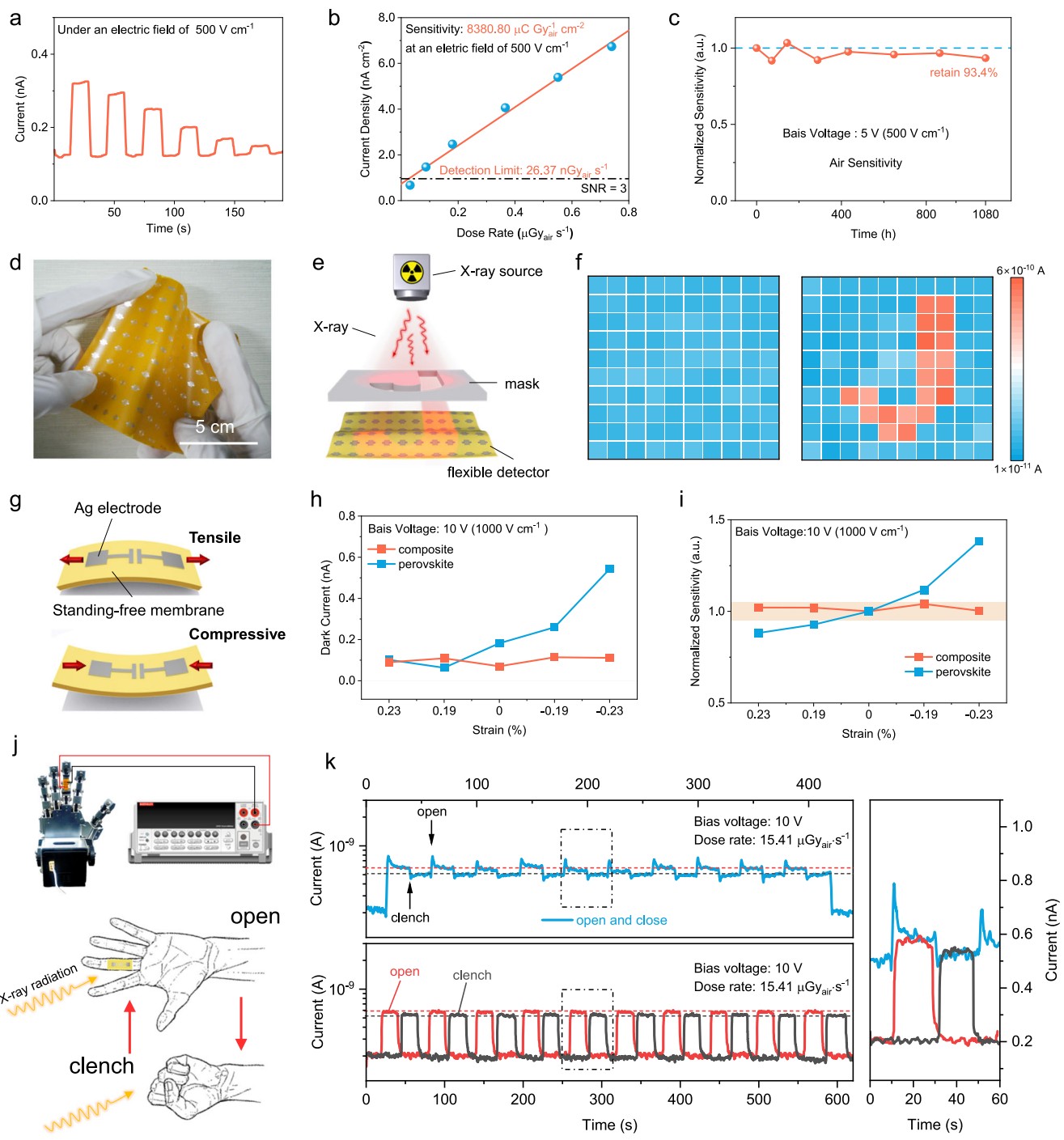

**Fig. 4 | X-ray detection performance and strain-insensitivity in wearable application. a** X-ray response of the champion detector based on composite membrane under dose rates ranging from 0.74 $\mu Gy_{air}\,s^{-1}$ to 0.03 $\mu Gy_{air}\,s^{-1}$ under a bias voltage of 5 V. **b** Current density as a function of incident dose rate of the X-ray detector. The horizontal dashed line is the noise current of the detector that corresponds to an SNR of 3. **c** Long-term stability of the composite membrane detectors (structure is glass/ITO/composite membrane/Ag) in ambient air with a relative humidity of 55%. **d** Photograph of a large-area flexible detector with a 10 × 10 pixel array. The scale bar is 5 cm. **e** Schematic illustration of the X-ray imaging process. **f** Dark current map and X-ray images with the letter "J" captured by the large-area flexible detector with a 10 × 10 pixel array. The X-ray image is obtained under an X-ray dose rate of 15.41 $\mu Gy_{air}\,s^{-1}$. **g** Schematic diagram of the bending direction of the flexible detectors. Dependence of (**h**) dark current and (**i**) normalized sensitivity on applied global strain of lateral devices based on pure perovskite membrane and composite membrane. **j** Photograph of a composite detector assembled on a robot hand for real-time monitoring of X-ray intensity. The robot hand underwent alternating gestures of open and clench. **k** Current–time curves of the X-ray detector under different conditions. The blue line was measured with the hand open or clenched while maintaining radiation. The red line and the black line were recorded under X-ray on-off conditions while the hand kept opening or clenching, respectively. Comparison of all response currents for open and clench movement and keeping states is shown in the right panel.

(Supplementary Fig. 23), immersion in water for 24 h (Supplementary Fig. 24), and ambient air storage 1080 h (Fig. 4c), both of which delivered negligible performance loss. With the synergy of the effective in-plane charge collection capacity and large-area

scalability of the composite membrane, we fabricated a 10 × 10-pixel array with interdigitated electrodes on account of a 100 cm² free-standing membrane for X-ray imaging (Fig. 4d, e). The recorded dark currents of each pixel are spatially uniform with the variation of

14% under an electric field of 1000 V cm$^{-1}$ (Fig. 4f). After exposure to X-ray together with a letter "J" metal mask, there is an abrupt augment in the photocurrents of unblocked pixels at a dose rate of 15.41 µGy$_{air}$ s$^{-1}$, generating a high-contrast image of letter "J". The integration of carrier transport capacity, scalability, and longevity in one composite membrane has fulfilled the technical requirements for radiation detection, and more critically, the involvement of ultraflexibility and elasticity could further extend its manifold application scenarios that conventional detectors can not easily achieve.

### Strain-insensitive and dynamic radiation monitoring

Since the perovskite phase stacked within PI layers undertakes considerably less strain under large mechanical deformation of the entire membrane, strain-insensitive devices that can function stably at wearable status are now possible. We measured the evolution of dark currents and sensitivities of pure perovskite and composite devices under different strain (Fig. 4g–i). The devices based on composite membranes show stable dark currents and almost invariant X-ray response sensitivities with <1.6% variation upon global strain of −0.23% – +0.23%, whereas the pure perovskite devices encounter variations of 19% in X-ray response sensitivities under the equivalent strain level.

To our knowledge, the strain-insensitive and dynamic detection have not yet been achieved in perovskite-based devices, because it is not possible to retain the electronic property in typical membranes under bending or twisting conditions. Our composite membranes can transfer load from perovskite to polymer phase, and thus allow us to accurately detect radiation under a dynamic wearable manner. The setup of the dynamic detection experiment is shown in Fig. 4j. Specifically, a flexible X-ray detector was closely mounted on the middle finger of a robot hand, and moved as it opened and clenched, while the current signal of the X-ray detector was real-time recorded by using a digital source meter. The static X-ray response currents at hand open or clenching states were firstly characterized to be 0.58 nA and 0.54 nA, respectively, the difference between which is caused by the shielding effect the manipulator itself (Fig. 4k). When releasing the detector under close state, i.e., from attached state to unattached state, the measured X-ray response current of 0.55 nA approached that of the attached one, which means that the readout signal by bended device can accurately reflect the radiation intensity irrespective of the physical deformation (Supplementary Fig. 25). Although the current variation between two actions was quite small, the periodic current fluctuation could be harvested when the hand repeatedly opens and closes, and the signal intensity is equal to the values measured at static state. Therefore, the strain-insensitive detector made the hand action discernible with respect to the current signal, and more importantly, permits the dynamic monitoring of radiation intensity in complicated environments in a wearable manner.

### Fatigue tolerance of the radiation detector

The success of composite membranes in ultraflexible detectors inspires us to explore their endurance to extreme fatigue tests. By using a film-on-elastomer configuration, we applied large compressive stress on both edges of the composite membrane to 67% of the initial length to induce micro-wrinkles on the surface (Fig. 5a, b)[51]. At the compressed state, wrinkled stripes with a radius <200 µm can be seen perpendicular to the strain direction. We found that there is no significant change in both photocurrent and dark current of the composite membrane device under 2000 cycles of severe consecutive deformation from the flat state to the wrinkled state (Fig. 5c). And the normalized sensitivity remains at 1.03 times the initial after 2000 cycles (Fig. 5d).

To further assess the mechanical robustness of the membrane, continuous deformation cycling tests were performed at a bending radius of 1.5 mm. With the layered microstructure design, the surface

and cross-sectional morphology of the membrane remains almost unchanged without any cracks during 10,000 bending cycles (Fig. 5f and Supplementary Fig. 26), while microcracks with radius of 380 nm generates and propagates vertically to the bending direction in the pure perovskite film after 2000 bending cycles (Supplementary Fig. 27). Moreover, the sensitivity of X-ray detection was recorded under the bias voltage of 10 V, and it almost keeps the initial value (97.4%) even after 30,000 cycles (Fig. 5g). Therefore, both the fatigue cycling experiments of wrinkling and repetitive bending did not degrade the device performance (Fig. 5e). It should be pointed out that the curvature radius of both fatigue tests (1.5 mm for bending, and <200 µm for wrinkling) employed in this work are much harsher than any reported ones undergone by perovskite device. We summarized the mechanical endurance under bending tests of different kinds of reported flexible devices in Fig. 5h, and the details about corresponding bending test parameters are shown in Supplementary Table 1. In fact, perovskite devices in previous studies have normally been subjected to R3-R10 standards (curvature radius = 3–10 mm), and only a few works have been subjected to R2 tests. Therefore, the layered microstructure of the membrane overcomes the intrinsic limit of perovskite materials in flexibility, so that it can be adapted to many extreme application scenarios.

## Discussion

In summary, we demonstrate a layered polymer-perovskite multi-material structure that converts the mechanically fragile perovskite into an ultra-flexible, fatigue-tolerant membrane without sacrificing the electronic performance. Through a scalable layer-by-layer spray-coating procedure, the PI and perovskite components with a suitable ratio can be assembled into a periodically layered structure of closely stacked bicontinuous phases, wherein the PI and perovskite function as strain-absorbing and semiconducting units, respectively. The connectivity and crystallinity of perovskite networks enable a high sensitivity of 8380.80 µC Gy$_{air}^{-1}$ cm$^{-2}$ and a low detection limit down to 26.37 nGy$_{air}$ s$^{-1}$. Using this approach, we resolved two previous limitations in the flexible perovskite-based devices that have rarely been achieved in previous studies: mechanical fatigue tolerance and insensitivity of electrical performance to deformation. Particularly, X-ray detectors based on the membrane have undergone 30,000 bending cycles under the extreme bending radius of 1.5 mm, and 2000 cycles of wrinkling tests (micro-wrinkle radius <200 µm) without notable performance loss. Furthermore, the insensitive photoresponse to deformation also permits the device to real-timely monitor the radiation signal under the dynamic mode, which would be useful for wearable applications. The fabrication protocol can be expanded to different types of perovskites, polymers, and substrates, which offers more possibilities for flexible perovskite-based electronics.

## Methods
### Materials
The following materials were used in the experiments: cesium bromide (CsBr; 99.5%, Macklin), lead bromide (PbBr$_2$; 99.0%, Macklin), methylammonium bromide (MABr; Greatcell Solar Materials), lead iodlde (PbI$_2$; 99.99%, Sigma-Aldrich), formamidinium iodide (FAI; Greatcell Solar Materials), silver bromide (AgBr; 98%, Aladdin), bismuth bromide (BiBr$_3$; 98%, Macklin), PVP (K30, Macklin), DMSO; 99.9%, Sigma-Aldrich, N, N-dimethylformamide (DMF; 99%, Alfa Aesar) and PI; 20% solid content, Yijiasuhua). All chemicals were used without further purification.

### Preparation of precursor solution
The CsPbBr$_3$ perovskite stock solution was prepared by mixing 0.5107 g of CsBr and 0.8808 g of PbBr$_2$ in 20 mL of DMF/DMSO solution (volume ratio = 7:3), and the PI stock solution was diluted

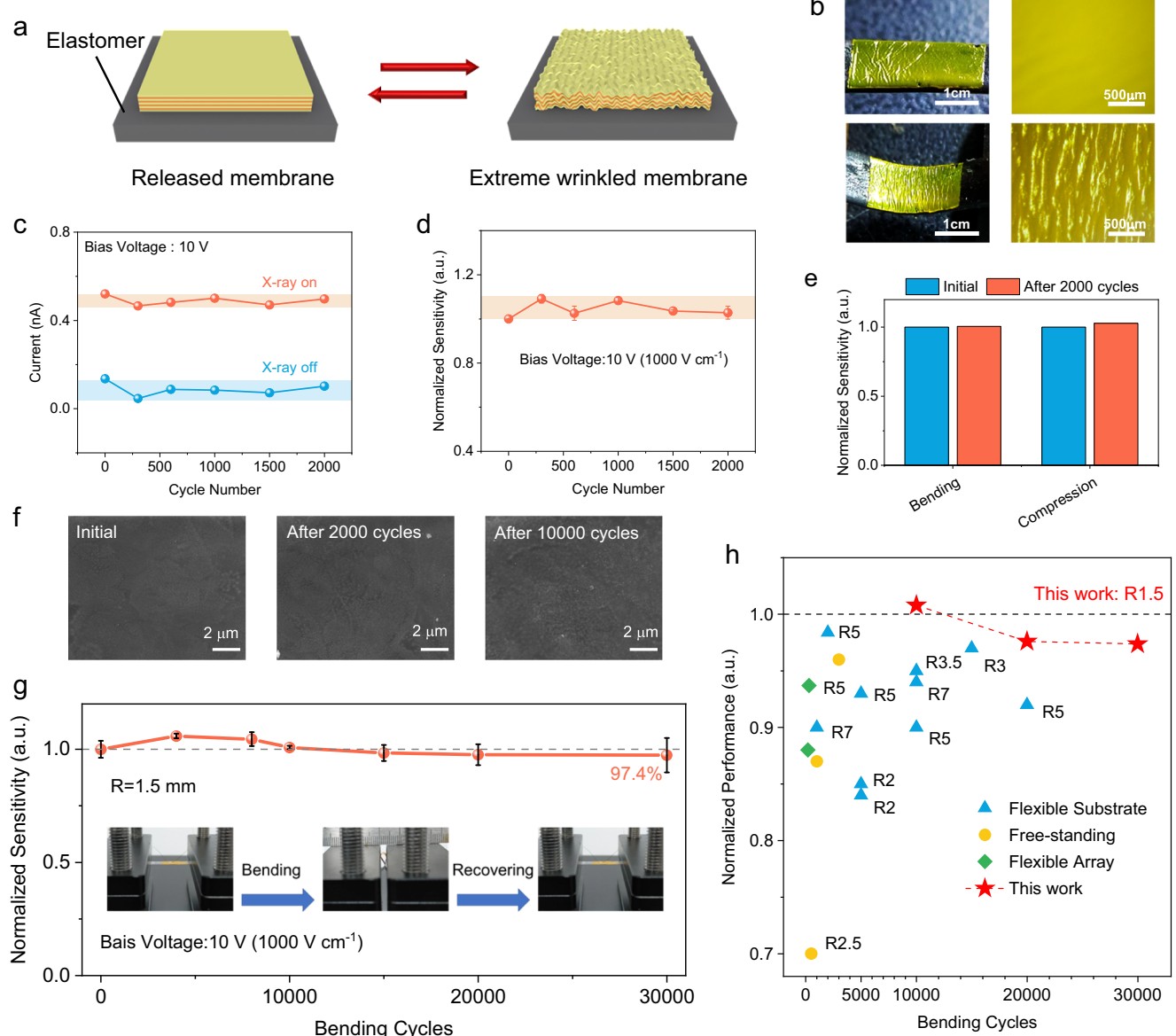

**Fig. 5 | Mechanical robustness under extreme deformation. a** Wrinkling experiment of the composite membrane attached to a soft elastomer. The device is compressed and relaxed repeatedly by stretching and releasing the elastomer. **b** Photographs (left) and optical microscope images (right) of released and wrinkled membrane under extreme stress membrane. Scale bars are 1 cm (left) and 500 μm (right). **c** On-off current and (**d**) normalized photocurrent of the ultraflexible X-ray detector upon 2000 compression cycles measured under the dose rate of 15.41 $\mu Gy_{air}$ $s^{-1}$ and bias voltage of 10 V. Error bars represent SD. **e** Comparison of normalized sensitivity before and after extreme deformation in

different modes of the flexible X-ray detectors. **f** Top-view SEM images of composite membranes after different bending cycles. Scale bars are 2 μm. **g** Normalized X-ray sensitivity versus bending cycles of the ultraflexible X-ray detectors measured under a bias voltage of 10 V. The bending radius is 1.5 mm for the stability tests. The inset shows the bending process of a composite device. Error bars represent SD. **h** Summary of the mechanical endurance of existing perovskite-based X-ray detectors under bending experiments. Details about references are shown in Supplementary Table 1.

with DMF solvent to a solid content of 15%. Then, both stock solutions were mixed to obtain mass fractions of perovskite between 10 and 90%. For MAPbBr$_3$ perovskite stock solution, 0.2687 g MABr, 0.8808 g PbBr$_2$ were dissolved in 1.6 mL of DMSO solution. For FAPbI$_3$ perovskite stock solution, 0.4265 g FAI and 1.1433 g PbI$_2$ were dissolved in 1.6 mL of DMF/DMSO solution (volume ratio = 4:1) and 0.0601 g of AgBr, 0.0681 g of CsBr and 0.1436 g of BiBr$_3$, corresponding to Cs$_2$AgBiBr$_6$ were dissolved in 1.6 mL of DMF/DMSO mixture solution (volume ratio = 4:1). Then, the PI stock solution of 15% solid content was added into each perovskite stock solution to obtain the precursor solution with 65% mass fraction of perovskite. For PVP-perovskite membrane, PVP powder was directly added into

the perovskite stock solution to obtain a 10% mass fraction of PVP. All the solutions were filtered with a 0.45 μm polytetrafluoroethylene membrane before use.

**Film and device fabrication**

Indium tin oxide-coated glass or PET was employed as the substrate for pure CsPbBr$_3$ perovskite films. The free-standing membrane was deposited on bare glass or a silicon wafer. All substrates were sequentially sonicated in acetone, ethanol, and water for 20 min each, followed by UV ozone treatment for 15 min. Then, the perovskite-based films were fabricated by the ultrasonic spray-coating technique. In a typical deposition process, the solution became aerosol droplets

with the assistance of an ultrasonic nebulizer (power, 50 W) and was carried to the nozzle with air as the gas carrier under 0.9 psi. The spray-head was programmed to move across the substrate at a speed of 10 mm s$^{-1}$ at a height of 30 mm, and the substrate was held at 150 °C in a single pass. The flow rate is 24 mL h$^{-1}$. The film needed to dry for 20 s at 150 °C before the next cycle. After deposition, the composite membrane can be removed from the glass substrate. Finally, 100 nm lateral Ag electrodes with channel length of 100 μm were thermally evaporated through a shadow mask for X-ray detection experiments. Additionally, interdigitated Ag electrodes with a channel length of 200 μm were thermally evaporated for 10 × 10 pixels array X-ray detection experiment.

### Structure characterization

The morphology of films was characterized by field emission scanning electron microscopy (FESEM, HITACHI S4800) and atomic force microscopy (AFM, NT-MDT). Transmission electron microscopy characterization was performed on a Thermo Fisher Talos F200× microscope under 200 kV. Aberration-corrected scanning transmission electron microscopy (AC-STEM) characterization was performed on a Thermo Fisher Talos F200× microscope under 200 kV (an aberration-corrected Nion UltraSTEM-100 operating at 60 kV). EDX was carried out using four in-column Super-X detectors. The crystallographic information was investigated by powder X-ray diffraction (Bruker Advance D8 X-ray diffractometer, Cu Kα radiation, 40 kV, 40 mA). Raman spectroscopy was recorded using a Renishaw InVia Reflex confocal Raman spectrometer with a NIR laser ($\lambda$ = 633 nm). Fourier-transform infrared spectra (FT-IR) were collected from a Thermo Nicolet 6700. Time of flight secondary ion mass spectroscopy (ToF-SIMS VI, IONTOF GmbH, Muenster) was used to probe the elemental distribution of perovskite films.

### Mechanical strength test

The hardness and modulus of the membranes were probed by the nanoindentation measurement (Bruker Hysitron TI 950) with a Berkovich indenter (three-sided pyramid shape tip) in a load-controlled mode of 1.5 mN. For the stress–displacement curve measurements, a double-cantilever beam delamination technique was used on an electronic universal testing machine (LGD500). The membrane sample was cut into 1 × 3 cm$^2$ rectangle. The specimen was mounted on the tensile tester fixture, and stretched at an operational rate of 0.5 mm min$^{-1}$.

### Photoelectronic characterization

Absorption spectra of PI and composite membranes were measured by using a Cary 500 UV–vis–NIR spectrophotometer. Time-resolved PL spectra were acquired using a Fluorolog-3-p spectrophotometer with the excitation wavelength of 528 nm. We fabricated a sandwich device with the structure Ag/active membrane/Ag. The membrane was fixed in PDMS and cut into strips, depositing Ag on both sides to form the lateral devices. For μτ product measurement, a 470 nm LED is used as excitation light, modulated at 50 Hz by a function generator. The photoconductivity current of the device was recorded by using a Keithley 2400 digital source meter. The resistivity of the device under dark conditions was recorded by a Keithley 4200 source meter.

### X-ray detection and imaging

During the X-ray detection experiment, the composite membranes were exposed to a Cu X-ray tube (Canon, A40) with a tube voltage of 40 kV. The dose rates were controlled by adjusting the tube current (2–40 mA) and aluminum foil, which was calibrated by RaySafe X2 R/F sensor. The X-ray response current was measured using a Keithley 2400 digital source meter. The sensitivity ($S$) can be calculated based on the equation:

$$S = \frac{I}{AD} \tag{1}$$

where $I$ is the response current, $A$ is the effective area. And the effective area was determined by a metal mask. The SNR can be calculated according to the equation:

$$SNR = \frac{J_{on} - J_n}{\sqrt{\frac{1}{N} \sum_i^N (J_i - J_{on})}} \tag{2}$$

where $J_{on}$ is the current density with X-ray irradiation, $J_n$ is the noise current density, and $J_i$ is the instantaneous value of $J_{on}$. A large-area array device constructed with 10 × 10 pixels was subjected to the X-ray imaging experiment. During the test, a metal mask was placed between the X-ray tube and the large-area X-ray device, and the current signal of each pixel at 10 V was recorded by a Keithley 2400 digital source meter to reproduce the image.

## Data availability

Source data are provided with this paper. The data within the Supplementary Information are available from the corresponding authors. Source data are provided with this paper.

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

## Acknowledgments

This work was financially supported by National Natural Science Foundation of China (22379044, 12304109, 52203330), Shanghai Pilot Program for Basic Research (22TQ1400100-5), Science and Technology Commission of Shanghai Municipality (23520710700), Key Program of the National Natural Science Foundation of China (22239001), the Fundamental Research Funds for the Central Universities (JKD01251505, JKVD1251041), Shanghai Engineering Research Center of Hierarchical Nanomaterials (18DZ2252400) and Shanghai Frontiers Science Center of Optogenetic Techniques for Cell Metabolism (Shanghai Municipal Education Commission).

## Author contributions

Y.L. designed and conducted the experiments. Y.Zheng carried out the theoretical simulation. C.Z., D.L., S.Z., and Z.W. helped with device performance measurement. Q.L., Y.Zhu, and M.L. contributed to the SEM characterization. Q.L. contributed to the Raman characterization. X.L. contributed to the FT-IR characterization. Y.L. and S.Y. wrote the manuscript. D.L., Y.P., and Y.Zheng assisted with manuscript revision. Y.H., S.Y., and H.G.Y. provided all support needed in this work. All authors contributed to the general discussion and reviewed the paper.

## Competing interests

The authors declare no competing interests.
