## [Transparent Peer Review file · Nature Communications]

Layered polymer-perovskite composite membranes for ultraflexible fatigue-tolerant optoelectronics

Corresponding Author: Professor Shuang Yang

Version 0:

Reviewer comments:

Reviewer #1

(Remarks to the Author)

This article reports on the design and fabrication of a periodic multilayered polymer-perovskite membrane, which achieves both plastic-like mechanical behaviors and semiconducting property. The manuscript is well prepared, and the results are interesting. Particularly, the flexible device shows a high X-ray detection sensitivity of $8380.8 \mu\text{C Gy}^{-1} \text{cm}^{-2}$, and maintains 97.4% of the initial detection sensitivity for 30000 bending cycles under a bending radius of 1.5 mm, which exceeds the performance of the state-of-the-art flexible perovskite detectors reported in the literature. We recommend that a revised article addressing the points below be considered for publication.

(1) This work employed a spraying process to achieve large-area preparation and preparation on both flat and shaped substrates, demonstrating advantages compared to other processes. The deposition process involves several repeated cycles of printing and annealing. However, such process may be compatible with other fabrication techniques, like spin coating or dip coating. Can these techniques also be used to fabricate layered membranes?

(2) For spray deposition, many parameters would affect the morphology and structure of the membrane. I suggested that the authors should elaborate on this.

(3) The cross-sectional SEM shows that the perovskite grains are completely encapsulated by the polymer, which contributes to the resistant to water erosion. And it shows a long-time stability for 1080 h with ambient air storage in Fig.4c. Is this composite membrane stable in water?

(4) In general, the layered structure tends to delaminate at the interfaces of these stacked laminates after repeated bending. The authors should supplement the relevant characterization to clarify this.

(5) The micro-wrinkle radius was mentioned in several places in the manuscript, but the units seem to be inconsistent. To avoid misunderstanding, the units should be confirmed by the authors.

(6) Supplementary Fig. 13 could be improved - the units of the wavenumber should be added in the figure.

Reviewer #2

(Remarks to the Author)

In this manuscript, Li et al. report a bicontinuous layered structure of polyimide and perovskite for achieving mechanical flexibility and electrically robustness simultaneously. The energy dissipation and electrical percolation mechanism were well characterized and analyzed, and it would be useful for flexible integration of perovskite-based electronics. I believe it could be of interest for the community. However, there are some concerns:

1. The authors mentioned that such layered structure is compatible with polymers, like PI and PVP. It would be important to explain which kind of polymers are suitable for fabricating this unique structure. The influential factors in the selection of polymers are recommend to be explored.

2. The composite membranes in this work can be simply peeled off from the glass substrate to form free-standing membranes. Is the glass substrate necessary for peeling off?

3. In Fig. 3b-c, the resistivity of the vertical devices is higher than that of the lateral devices at the same perovskite mass

fraction. However, it shows in the Fig. 3g that the resistance of the lateral devices is higher than that of the vertical devices. The authors should provide detailed explanation. And the manuscript lacks the detailed description about Fig. 3g.

4. In mechanical analysis section, the energy dissipation mechanism was ascribed to the elastic deformation of polymer phase under bending. As the perovskite phase is fragile, plastic deformation may happen during the bending test. More experimental characterizations may give further evidence.

5. The calculation details about the anisotropic resistivity ratio in Supplementary Fig. 17 have not been given. The authors should explain the link between the optimal perovskite mass fraction and the anisotropy ratio.

6. There are some errors in the description. In the description of Fig. 3f, the difference of the derived mobility-lifetime ($\mu\tau$) products between the lateral device ($1.04 \times 10^{-4} \text{ cm}^2 \text{ V}^{-1}$) and vertical device ($2.78 \times 10^{-7} \text{ cm}^2 \text{ V}^{-1}$) is three orders of magnitude instead of two.

Version 1:

Reviewer comments:

Reviewer #1

(Remarks to the Author)

The authors addressed my concerns. I have no further concern.

Reviewer #2

(Remarks to the Author)

the revision should be changed enough for publication in this journal.

Reviewer #1 (Remarks to the Author):

This article reports on the design and fabrication of a periodic multilayered polymer-perovskite membrane, which achieves both plastic-like mechanical behaviors and semiconducting property. The manuscript is well prepared, and the results are interesting. Particularly, the flexible device shows a high X-ray detection sensitivity of $8380.8 \mu\text{C Gy}^{-1} \text{cm}^{-2}$, and maintains 97.4% of the initial detection sensitivity for 30,000 bending cycles under a bending radius of 1.5 mm, which exceeds the performance of the state-of-the-art flexible perovskite detectors reported in the literature. We recommend that a revised article addressing the points below be considered for publication.

Comment 1: This work employed a spraying process to achieve large-area preparation and preparation on both flat and shaped substrates, demonstrating advantages compared to other processes. The deposition process involves several repeated cycles of printing and annealing. However, such process may be compatible with other fabrication techniques, like spin coating or dip coating. Can these techniques also be used to fabricate layered membranes?

Response: We really appreciate the reviewer's recognition and positive comments on our work. We agree with the reviewer that the coating process may be compatible with other fabrication techniques. To enable a direct comparison of different processes, we conducted spin-coating experiments with similar processing condition, which involves spin-coating procedures of perovskite-polyimide precursor solution at 3000 rpm for 30 s and then annealed at 150 °C for 20 s before the next cycle. Cross-sectional scanning electron microscope (SEM) images of the membranes coated by different cycles are shown in Fig. R1. Single spin-coating resulted in the polymer-perovskite composite membranes, in which the perovskite crystals concentrated in the middle of the membrane. However, the vertical distribution of perovskite crystals by spin-coating was relatively discrete compared with the spray printing one. These results indicate the very similar crystallization process of precursor solution upon heating. The emergence of the layered structure is attributed to the inherently distinct physicochemical processes experienced by the perovskite and polymer constituents during solvent evaporation, which corroborates the analysis presented in the manuscript.

To further investigate the layer formation process, up to five spin-coating cycles were conducted. Notably, even with annealing step of 20 s before each coating step, the previously deposited composite membranes exhibited poor resistance to solvent-

induced redissolution, a phenomenon not observed in the spray-coating. Cross-sectional SEM analysis further reveals that the composite membrane fabricated via five spin-coating cycles lacks a distinct layered structure. This can be attributed to two possible reasons: (1) the vertical distribution of perovskite crystals is not narrow enough by spin-coating, and (2) the aerosolized solvent generated by the ultrasonic nebulizer during spray-coating promotes rapid evaporation and reduces the likelihood of membrane redissolution. These results highlight the importance of spray coating for the fabrication of multilayered polymer-perovskite membrane.

The discussion of this part has been added in the revised manuscript (lines 104-107) and the revised Supplementary Information (Supplementary Fig. 5).

Fig. R1 | Cross-sectional SEM images of composite membranes via spin-coating for single and five cycles. Scale bars are 1 μm.

Comment 2: For spray deposition, many parameters would affect the morphology and structure of the membrane. I suggested that the authors should elaborate on this.

Response: We really thank the reviewer's valuable comments and suggestions. Following the reviewer's suggestion, we investigated the influence of different parameters on the membrane structure through controlled variable experiments. In contrast to other one-step fabrication process, spray-coating process involves aerosols to the surface of the membrane, and at the same time, the perovskite precursor generally solidifies from the bottom interface¹. Computational fluid dynamics (CFD) studies based on the principle of fluid mechanics² have been conducted to reveal the mechanism of the spray-coating process. To establish a dynamic equilibrium among the aerosol, liquid, and solid phases, key spray-coating parameters—including air pressure, spray speed, flow rate, nozzle height, and travel speed—have primarily been optimized to ensure an appropriate amount of aerosolized solution reaches the substrate, thereby

enabling uniform membrane deposition.

In addition to these variables, other factors including annealing temperature, annealing duration, and the number of spray cycles also significantly influence the structural characteristics of the composite membrane. To systematically evaluate their effects, we conducted a series of comparative experiments by varying the annealing temperature (100 °C, 150 °C, and 170 °C), annealing time (20 s and 2 min), and the number of spray cycles (15 and 25 cycles), respectively. Cross-sectional SEM images reveal that perovskite grain size decreases with increasing annealing temperature (Figs. R2a-c). At low temperature (100 °C), the slow crystallization of chalcocite may result in the loss of the layered structure. Conversely, annealing at high temperature (170 °C) reduces interlayer adhesion and induces void formation, similar to the effect observed in the sample annealed for a longer period (Fig. R2d). By optimizing annealing temperature and duration, precise thickness control can be achieved through cycle number adjustment (Fig. R2e).

The additional discussion of this part has been added in the revised manuscript (lines 124-127) and the revised Supplementary Information (Supplementary Fig. 7).

Fig. R2 | Cross-sectional SEM images of composite membranes. Spray for 15 cycles and anneal for 20 s at different anneal temperatures: (a) 100 °C, (b) 150 °C and (c) 170 °C. d, Spray for 15 cycles and anneal for 2 min at 150 °C. e, Spray for 25 cycles and anneal at 150 °C for 20 s. Scale bars are 20 μm.

Comment 3: The cross-sectional SEM shows that the perovskite grains are completely encapsulated by the polymer, which contributes to the resistant to water erosion. And it shows a long-time stability for 1080 h with ambient air storage in Fig.4c. Is this composite membrane stable in water?

Response: We are grateful for the insightful suggestions. To further evaluate the stability of the composite membrane, we conducted additional experiments focused on its water stability, as suggested. The membrane was immersed in deionized water for 24 h (Fig. R3a). After drying, no surface whitening—typically associated with the hydrolysis of CsPbBr₃—was observed, indicating the absence of significant degradation. This observation was further corroborated by the XRD patterns (Fig. R3b), which revealed no notable structural changes. Moreover, the X-ray detection sensitivity of the immersed membrane retained 98.8% of the initial one (Fig. R3c), further confirming its excellent water stability.

The supplementary experiments have been incorporated into the revised manuscript (lines 299-302) and the revised Supplementary Information (Supplementary Fig. 24).

Fig. R3 | Water stability of composite membranes. (a) Photographs, and (b) XRD patterns of composite membranes before and after immersion in water for 24 h. c, Current density as a function of incident dose rate of the X-ray detectors.

Comment 4: In general, the layered structure tends to delaminate at the interfaces of these stacked laminates after repeated bending. The authors should supplement the relevant characterization to clarify this.

Response: We appreciate this valuable comment. Accordingly, cross-sectional SEM images of the composite membrane before and after 10,000 bending cycles ($R = 1.5$

mm) were operated. We found that no obvious interfacial delamination happened, indicative of strong interaction between the polymer and perovskite phases (Fig. R4). This is consistent with the mechanical stability results of flexible detectors.

And the investigation has been addressed in the revised manuscript (lines 355-360) and the revised Supplementary Information (Supplementary Fig. 26).

Fig. R4 | Cross-sectional SEM images of composite membranes before and after 10,000 times bending. Scale bars are 10 μm .

Comment 5: The micro-wrinkle radius was mentioned in several places in the manuscript, but the units seem to be inconsistent. To avoid misunderstanding, the units should be confirmed by the authors.

Response: We are grateful for the suggestion to check units of the micro-wrinkle radius. To avoid any misunderstanding, we carefully examined the unit of the micro-wrinkle radius and corrected the erroneous value of "200 nm" to "200 μm " in the revised manuscript (lines 363-366).

Comment 6: Supplementary Fig. 13 could be improved - the units of the wavenumber should be added in the figure.

Response: We appreciate the reviewer's careful checking of the formats. The guidance on adding the units of the wavenumber has been carefully followed (Fig. R5) and update to the revised Supplementary Information (Supplementary Fig. 15).

Fig. R5 | FT-IR spectra of pure polyimide (PI) and composite membranes.

Reviewer #2 (Remarks to the Author):

In this manuscript, Li et al. report a bicontinuous layered structure of polyimide and perovskite for achieving mechanical flexibility and electrically robustness simultaneously. The energy dissipation and electrical percolation mechanism were well characterized and analyzed, and it would be useful for flexible integration of perovskite-based electronics. I believe it could be of interest for the community. However, there are some concerns:

Comment 1: The authors mentioned that such layered structure is compatible with polymers, like PI and PVP. It would be important to explain which kind of polymers are suitable for fabricating this unique structure. The influential factors in the selection of polymers are recommend to be explored.

Response: We really appreciate the reviewer's positive comments and valuable suggestions. Following the reviewer's advice, we summarized the key factors influencing the selection of polymers for the multilayered composite membranes:

(1) Miscibility with the perovskite precursor solution: If the solvent is poor for the polymer, premature phase separation may occur during solvent evaporation³⁻⁵. In this work, a DMF/DMSO solvent system was used; therefore, the selected polymer must be soluble in this solvent mixture to ensure adequate mixing with the perovskite and enable uniform spray-coating.

(2) Gelation capability: Not all polymer solutions undergo gelation prior to solvent removal. Gelation behavior depends on several factors, including polymer concentration, solvent compatibility, and polymer structure. Typically, low-concentration polymer solutions or polymers with a high content of rigid segments tend to crystallize during solvent removal rather than form a gel network⁶. For example, polyaniline tends to crystallize rather than gelate under such conditions^{7,8}.

The polymers used in this work, e.g., polyimide (PI) and polyvinyl pyrrolidone (PVP), meet the above requirements and can be assembled into layered polymer-perovskite structures via a layer-by-layer spray-coating process.

This issue has been refined in the revised manuscript (lines 86-89).

Comment 2: The composite membranes in this work can be simply peeled off from the glass substrate to form free-standing membranes. Is the glass substrate necessary for

peeling off?

Response: We thank the reviewer for the comments. In fact, glass is one type of substrates that can be delaminated from the composite membranes. In our studies, aluminum alloy has been used as the substrate for preparing membranes with unique shapes, and the results are shown in Fig. R6. We also confirmed that PDMS can serve as a peelable substrate. Based on the results, the following criteria are suggested for enabling composite membranes delamination:

(1) A smooth substrate surface, which suppresses mechanical interlocking.

(2) Weak interfacial adhesion between the substrate and the composite film, which must be lower than the membrane's intrinsic cohesive strength. The polyimide matrix possesses strong intermolecular cohesion, primarily through both physical and chemical cross-link⁹⁻¹¹. In contrast, the interaction between the membrane and the substrate is dominated by weak, non-covalent forces (e.g., van der Waals forces and hydrogen bonds), as no covalent bonding is formed at the interface. This disparity in interaction strength favors interfacial separation over cohesive failure within the membrane during the peeling process.

(3) A mismatch in coefficients of thermal expansion (CTE) between the substrate and the membrane would induce interfacial stress and weaken interfacial contacts, which is beneficial for interfacial delamination¹².

Fig. R6 | **a**, Schematic diagram of spray-coating process on the aluminum alloy substrate. **b**, Photographs of composite membranes peeled from the aluminum alloy

substrates. Scale bars are 5 cm.

Comment 3: In Fig. 3b-c, the resistivity of the vertical devices is higher than that of the lateral devices at the same perovskite mass fraction. However, it shows in the Fig. 3g that the resistance of the lateral devices is higher than that of the vertical devices. The authors should provide detailed explanation. And the manuscript lacks the detailed description about Fig. 3g.

Response: We thank the reviewer for the valuable comment. Figs. R7a and b present the variation in resistivity of the composite membranes in both the lateral and vertical directions with increasing perovskite mass fraction. These data highlight the intrinsic resistive behavior, demonstrating that the charge transport properties can be effectively tuned by adjusting the perovskite content. Fig. R7c, on the other hand, displays the time-resistance curves of the lateral and vertical devices. The measurements reflect not only the resistivity of the composite membranes, but also the influence of device geometry (i.e., cross-sectional area and length). The higher resistance observed in the lateral device suggests a lower dark current under dark conditions, which is beneficial for detection applications.

We have incorporated this explanation into the revised manuscript (lines 270-273).

Fig. R7 | Structure dependent anisotropic electrical transport. Resistivities of (a) lateral and (b) vertical devices as a function of the mass fraction of perovskite. Insets show the configuration of the measured devices. Membranes maintain lamellar structure in blue area. Error bars represent the standard deviation (SD). c, Resistance measurements of lateral and vertical devices with the perovskite mass fraction of 65%.

Comment 4: In mechanical analysis section, the energy dissipation mechanism was ascribed to the elastic deformation of polymer phase under bending. As the perovskite phase is fragile, plastic deformation may happen during the bending test. More

experimental characterizations may give further evidence.

Response: We thank the reviewer for the valuable comments. Cross-sectional SEM studies were performed on the composite membrane after 10,000 bending cycles ($R = 1.5$ mm) in Fig. R8a. No evident plastic deformation was observed in the perovskite layer. As shown in the magnified SEM image (Fig. R8b), the perovskite grains still retain the granular shape; polymer material also remains to be continuous between the grains, acting as a mechanical buffer¹³. Therefore, there was no notable plastic deformation in the membrane after bending tests. This can be explained by that the strain upon bending were primarily transferred to the soft polymer phase in this layered structure^{14–16}.

And the experimental characterization has been addressed in the revised manuscript (lines 355-360) and the revised Supplementary Information (Supplementary Fig. 26).

Fig. R8 | **a**, Cross-sectional SEM images of composite membranes before and after bending for 10,000 times. Scale bars are 10 μm. **b**, the magnified SEM image of composite membranes. Scale bar is 2.5 μm.

Comment 5: The calculation details about the anisotropic resistivity ratio in Supplementary Fig. 17 have not been given. The authors should explain the link between the optimal perovskite mass fraction and the anisotropy ratio.

Response: We thank the reviewer for the careful checking of our work. We have clarified the concept of the anisotropy ratio as “resistivity ratio of the vertical device to the lateral device at the same mass fraction,” and have included this definition in the revised manuscript (lines 264-267).

As illustrated in Fig. R9a, an increase in the mass fraction of the filler (perovskite) leads to enhanced photoelectronic property of the composite film, accompanied by a reduction in flexibility. Therefore, the determination of the optimal perovskite mass fraction requires a reconciliation between photoelectronic property and flexibility, which is achieved within the bicontinuous phases.

According to the aforementioned definition, the anisotropy ratio reflects the relative conductivity in the lateral and vertical directions. As shown in Fig. R9b, the curve can be divided into four regions: (1) When the filler fraction $< 35\%$, the composite is non-conductive in both directions, and the anisotropy ratio remains unchanged with increasing mass fraction. (2) With filler fraction of $35\% \sim 50\%$, the lateral direction reaches its critical threshold first, while the vertical direction remains non-conductive. The rising anisotropy ratio in this region is mainly due to enhanced lateral conductivity. The composite membrane exhibits good photoelectronic property while retaining flexibility. (3) For filler fraction of approximately 50% , the vertical direction reaches its critical threshold, corresponding to the peak in anisotropy ratio. In the range of $50\text{--}80\%$, vertical conduction improves obviously, while the bicontinuous structure of the perovskite and polymer phases is still preserved. At this stage, the composite membrane remains flexible, although its flexibility decreases with increasing filler content. (4) When the mass fraction exceeds 80% , the polymer phase loses its continuity, leading to the disappearance of anisotropy. As a result, the composite membrane becomes mechanically rigid. Therefore, a mass fraction of 65% was selected as the optimal value, as it represents a balance between enhanced photoelectronic property and acceptable flexibility.

The detail discussion of this part has been supported in the revised Supplementary Information (Supplementary Fig. 19).

Fig. R8 | **a**, Dependence of mechanical flexibility and photoelectric property of the composite membrane as a function of filler (perovskite) mass fraction (f). **b**, Electrical anisotropic ratio of the composite membranes with different mass fractions of perovskite derived from the lateral and vertical resistivities.

Comment 6: There are some errors in the description. In the description of Fig. 3f, the

difference of the derived mobility-lifetime ($\mu\tau$) products between the lateral device ($1.04 \times 10^{-4} \text{ cm}^2 \text{ V}^{-1}$) and vertical device ($2.78 \times 10^{-7} \text{ cm}^2 \text{ V}^{-1}$) is three orders of magnitude instead of two.

Response: We gratefully acknowledge the reviewer for the careful examination of our work. We have revised the description of ‘which is two orders of magnitude larger than the vertical one’ to ‘which is approximately three orders of magnitude larger than the vertical one’ in the manuscript (lines 273-275).

References

1. Qian, W. *et al.* An aerosol-liquid-solid process for the general synthesis of halide perovskite thick films for direct-conversion X-ray detectors. *Matter* **4**, 942–954 (2021).
2. Feng, X. *et al.* Spray-coated perovskite hemispherical photodetector featuring narrow-band and wide-angle imaging. *Nat. Commun.* **13**, 6106 (2022).
3. Luebbert, C., Real, D. & Sadowski, G. Choosing appropriate solvents for ASD preparation. *Mol. Pharmaceutics* **15**, 5397–5409 (2018).
4. Dohrn, S. *et al.* Solvent influence on the phase behavior and glass transition of amorphous solid dispersions. *Eur. J. Pharm. Biopharm.* **158**, 132–142 (2021).
5. Dohrn, S. *et al.* Phase behavior of pharmaceutically relevant polymer/solvent mixtures. *Int. J. Pharm.* **577**, 119065 (2020).
6. Cheng, H. *et al.* Supercooling and solvent depletion-driven poly(vinyl alcohol) film formation mechanism as elucidated by in situ synchrotron radiation X-ray scattering. *Macromolecules* **57**, 1569–1580 (2024).
7. Verma, D. & Dutta, V. Novel microstructure in spin coated polyaniline thin films. *J. Phys.: Condens. Matter* **19**, 186212 (2007).
8. Zhang, T. *et al.* Engineering crystalline quasi-two-dimensional polyaniline thin film with enhanced electrical and chemiresistive sensing performances. *Nat. Commun.* **10**, 4225 (2019).
9. Vanherck, K., Koeckelberghs, G. & Vankelecom, I. F. J. Crosslinking polyimides for membrane applications: A review. *Prog. Polym. Sci.* **38**, 874–896 (2013).
10. Duthie, X. *et al.* Thermal treatment of dense polyimide membranes. *J. Polym. Sci., Part B: Polym. Phys.* **46**, 1879–1890 (2008).
11. Hasegawa, M., Mita, I., Kochi, M. & Yokota, R. Charge-transfer emission spectra of aromatic polyimides. *J. Polym. Sci. C Polym. Lett.* **27**, 263–269 (1989).
12. Xue, D.-J. *et al.* Regulating strain in perovskite thin films through charge-transport layers. *Nat. Commun.* **11**, 1514 (2020).
13. Wu, Y. *et al.* In situ crosslinking-assisted perovskite grain growth for mechanically robust flexible perovskite solar cells with 23.4% efficiency. *Joule* **7**, 398–415 (2023).
14. Podsiadlo, P. *et al.* Ultrastrong and stiff layered polymer nanocomposites. *Science* **318**, 80–83 (2007).
15. Liu, X. *et al.* Topoarchitected polymer networks expand the space of material properties. *Nat. Commun.* **13**, 1622 (2022).

16. Chen, K. *et al.* Graphene oxide bulk material reinforced by heterophase platelets with multiscale interface crosslinking. *Nat. Mater.* **21**, 1121–1129 (2022).